# Cardiovascular Calcification as a Marker of Increased Cardiovascular Risk and a Surrogate for Subclinical Atherosclerosis: Role of Echocardiography

**DOI:** 10.3390/jcm10081668

**Published:** 2021-04-13

**Authors:** Andrea Faggiano, Gloria Santangelo, Stefano Carugo, Gregg Pressman, Eugenio Picano, Pompilio Faggiano

**Affiliations:** 1Fondazione IRCCS Ca’ Granda Ospedale Maggiore Policlinico, University of Milan, 20122 Milan, Italy; andreafaggiano95@gmail.com (A.F.); stefano.carugo@unimi.it (S.C.); 2San Paolo Hospital, Division of Cardiology, Department of Health Sciences, University of Milan, 20144 Milan, Italy; gloriasantangelo1982@gmail.com; 3Division of Cardiology, Heart and Vascular Institute, Einstein Medical Center, Philadelphia, PA 19141, USA; pressmang@einstein.edu; 4CNR, Institute of Clinical Physiology, Biomedicine Department, 56124 Pisa, Italy; picano@ifc.cnr.it; 5Fondazione Poliambulanza, Cardiovascular Disease Unit, University of Brescia, 25124 Brescia, Italy

**Keywords:** subclinical atherosclerosis, cardiac calcification, risk reclassification

## Abstract

The risk prediction of future cardiovascular events is mainly based on conventional risk factor assessment by validated algorithms, such as the Framingham Risk Score, the Pooled Cohort Equations and the European SCORE Risk Charts. The identification of subclinical atherosclerosis has emerged as a promising tool to refine the individual cardiovascular risk identified by these models, to prognostic stratify asymptomatic individuals and to implement preventive strategies. Several imaging modalities have been proposed for the identification of subclinical organ damage, the main ones being coronary artery calcification scanning by cardiac computed tomography and the two-dimensional ultrasound evaluation of carotid arteries. In this context, echocardiography offers an assessment of cardiac calcifications at different sites, such as the mitral apparatus (including annulus, leaflets and papillary muscles), aortic valve and ascending aorta, findings that are associated with the clinical manifestation of atherosclerotic disease and are predictive of future cardiovascular events. The aim of this paper is to summarize the available evidence on clinical implications of cardiac calcification, review studies that propose semiquantitative ultrasound assessments of cardiac calcifications and evaluate the potential of ultrasound calcium scores for risk stratification and prevention of clinical events.

## 1. Introduction

Cardiovascular (CV) diseases, primarily ischemic heart disease (IHD) and stroke, are still the leading cause of global mortality and a major contributor to disability. Indeed, prevalent cases of total CV diseases nearly doubled from 271 million in 1990 to 523 million in 2019, and the number of CV deaths steadily increased from 12.1 million in 1990, reaching 18.6 million in 2019 [1]. Because CV risk is the result of the interaction of different traditional risk factors (TRFs), in terms of severity and time to exposure, risk prediction algorithms combining multiple TRFs have gained a central role in CV disease prevention [2].

### 1.1. Current Cardiovascular Risk Prediction: Algorithms and Limitations

The European Society of Cardiology (ESC) SCORE charts [3], the American College of Cardiology/American Heart Association Atherosclerotic Cardiovascular Disease (ACC/AHA ASCVD) Risk Calculator [4] and the Framingham Risk Score (FRS) [5] are the main currently used risk algorithms, and all of them estimate the absolute risk of CV events over 10 years. They identify subjects at very high risks if the calculated SCORE risk is ≥10%, subjects at high CV risk if the SCORE risk is between 5% and 10%, the FRS risk is ≥20% or the ASCVD is ≥20%, subjects at moderate risk if the SCORE is between 1% and 5%, the FRS is between 10% and 20% or the ASCVD is between 7.5% and 19.9% [6,7].

Although algorithms are aimed to customize risk, they have at least three well-acknowledged limitations. First, they are primarily influenced by age; therefore, an asymptomatic young person with multiple TRFs is likely to be identified as low risk, while, conversely, an elderly person is likely to be identified as high risk regardless of actual risk [4,6]. Second, risk charts do not take into account either the time of risk exposure or some important TRFs such as family history for CV disease, obesity and glucose intolerance. They also miss important nontraditional risk factors, such as cumulative exposure to air pollution [8]. Finally, as postulated by the epidemiologist Geoffrey Rose, most CV events will develop in low-risk subjects, simply because they are much more numerous than those at high risk [9]. As a consequence, people with low estimated risk but who already have subclinical atherosclerosis could be underrecognized and undertreated. Therefore, it is recommended that asymptomatic people with calculated CV risks near the decisional thresholds (between two risk classes) undergo noninvasive testing (looking for subclinical atherosclerosis) for further risk stratification and possible reclassification [6]. For example, a recent manuscript suggests implementing the screening for subclinical atherosclerosis in subjects with a calculated SCORE ≥ 3 or a calculated Framingham ≥ 10 [10].

Reclassification means changing the risk class of a subject after complementing risk algorithms with other data; for example, an imaging technique [11]. Obviously, reclassification to a higher risk class can entail the adoption of preventive strategies, such as pharmacological treatment (i.e., statins); vice versa, a subject reclassified as at low risk of CV events does not require such preventive strategies. Anyway, reclassification can be considered appropriate (when “moving” individuals who will develop future CV events into higher estimated-risk levels) or not appropriate (when “moving” individuals who will develop future CV events into lower estimated-risk levels) [2]. Generally, a new risk prediction model is considered useful if it leads to a “net reclassification improvement (NRI)” of at least 10%, i.e., at least 10% of people are more appropriately reclassified with the new method compared with the old one [12].

### 1.2. The Role of Imaging in Detecting Subclinical Atherosclerosis and Reclassify Patients Risk

It is increasingly clear that the use of preventive models based on the detection of subclinical atherosclerosis and organ damage is useful [2,13]. Indeed, as Shah [14] emphasized, despite the lack of randomized clinical trial evidence, the totality of observational evidence supports “imaging-guided prediction and management” because:Detecting atherosclerosis (the consequences of which we aim to avoid) is better than simply identifying TRF exposure;Reclassification of low-risk subjects into higher-risk strata may guide appropriate therapy;Disease visualization might improve adherence to risk-modifying interventions by increasing awareness.

Several imaging modalities have been evaluated and proposed for the identification of preclinical atherosclerosis and phenotypic evidence of disease [15]. There are currently two primary imaging techniques used to reclassify low-intermediate cardiovascular risk subjects.

#### 1.2.1. Two-Dimensional (2D) Ultrasound of Carotid Arteries

This imaging modality allows the physician to detect both the presence of increased carotid intima-media thickness (CIMT) and carotid plaques [16]. Conflicting results have been published on the added value of CIMT measurements in cardiovascular risk prediction [17,18]. In fact, while Framingham investigators [19] showed that the maximal CIMT of the internal carotid artery added a modest value (NRI: 7.6%) in risk prediction, a meta-analysis [20] of about 46,000 patients showed that the addition of common CIMT measurements to the FRS was associated only with a small improvement (NRI 0.8%) in the 10-year risk prediction of myocardial infarction or stroke. However, carotid ultrasound can give information besides CIMT. The presence of carotid plaques conferred a two-fold increase in the risk of future CV adverse events (hazard ratio (HR): 2.3) in a population free of overt CV disease [21]. A meta-analysis by Peters et al. [22] also noted that the presence of carotid plaques added value for screening asymptomatic subjects at intermediate risk, improving the NRI from 8 to 11%. Therefore, maximum carotid plaque thickness seems to be a simple useful measure to enhance the prediction (NRI: 17.8%) of future cardiovascular disease events [23]. Furthermore, the morphological characteristics of carotid plaques are associated with future cerebrovascular ischemic events as well [24]. Indeed, a recent study conducted with magnetic resonance showed that in patients with carotid plaque the presence of intraplaque hemorrhage is a stronger predictor of stroke than any known clinical risk factors [25].

#### 1.2.2. Coronary Artery Calcium Score (CACs)

Coronary calcium detection by computed tomography (CT) is the most commonly used technique for the detection of subclinical disease, prognostic stratification of asymptomatic individuals and implementation of preventive strategies [26]. It is possible to quantitatively assess coronary calcium using Agatston CACs, a surrogate for plaque burden that has been shown to provide powerful prognostic information in multiple studies involving both sexes and multiple ethnic groups [14,27]. The body of evidence on the predictive and risk-reclassification role of CACs is large and founded on several well-designed prospective studies. It is now supported by a large amount of data the fact that the presence of coronary artery calcium provides independent incremental information in addition to TRF to predict all-cause mortality, whereas its absence (CACs = 0) identifies a group of asymptomatic subjects at very low CV risk (regardless of the presence of underlying risk factors) with a consequent reduced need for aggressive therapy or further diagnostic tests [22,28,29]. The risk class in which this technique has proved most useful is the intermediate one, in which adding CACs to TRFs resulted in a NRI of 55% [30]. Indeed, in the 6814 participants from the Multiethnic Study of Atherosclerosis (MESA) [30], CACs assessment allowed the reclassification of 292 subjects (16%) from moderate to high risk and of 712 (39%) from moderate to low risk. Furthermore, a recent analysis of the MESA data found that among persons with LDL-C ≥ 190 mg/dL (therefore at increased CV risk), a CAC of 0 was associated with a low risk of cardiovascular events, suggesting the utility of CACs for redefining risk in this patient group as well [31]. In support of this imaging technique, a brilliant Danish study [32] recently showed that coronary plaque burden, not stenosis per se, is the main predictor of CV events and death. Thus, patients with a comparable calcium burden measured as CACs generally have a similar risk for CV events regardless of whether they have nonobstructive or obstructive coronary artery disease. It should be emphasized that coronary CT angiography (CCTA), an imaging technique currently not recommended for reclassifying asymptomatic subjects, is, however, capable of detecting “soft” noncalcified plaque in patients who have CACs = 0 [33].

However, both of these techniques have limitations to consider. Regarding the 2D ultrasound imaging of the carotid arteries, despite providing a simple noninvasive (and relatively inexpensive) modality for detecting subclinical atherosclerosis, results show more subjects moving from intermediate-risk to lower-risk categories than to higher-risk categories. This implies the possibility of not undertaking preventive interventions in subjects who could benefit from them [34]. The major drawback of CT imaging is the exposure to ionizing radiation, which is particularly undesirable in young subjects, especially women. Its use is also limited by cost, by the impossibility of being performed “bedside” and by the limited availability in nonspecialized centers. CT also requires interpretation by a physician with specialized training (e.g., radiologist) who is often not the clinician taking care of the patient. This fact inevitably entails greater time, organizational and bureaucratic cumbersome, with the possibility that both the clinician and the subject/patient are discouraged from carrying out an in-depth diagnostic pathway to reclassify the CV risk.

Given these considerations, we would argue for the use of routine echocardiography as a practical technique in the detection of subclinical atherosclerosis. In this review, we discuss the utility of ultrasound assessment of cardiac calcifications as a tool to identify the presence of coronary atherosclerosis and improve risk stratification of asymptomatic subjects.

## 2. Definition and Epidemiology of Cardiac Calcifications

Cardiac calcifications are frequently encountered on routine echocardiographic examination or CT scanning. They are generally asymptomatic, and their prevalence varies according to the site evaluated, age and presence of cardiovascular risk factors (including chronic kidney disease [35] or diabetes [36]). The sites affected most often are the aortic valve (prevalence about 24%) and the mitral annulus (prevalence 8% and up to 15% with increasing age and number of risk factors) [37,38]. Furthermore, calcifications can often involve both the aortic and mitral valves simultaneously, especially in diabetic patients. Rossi et al. [39] observed that approximately 45% of 900 type-2 diabetic subjects had aortic valve sclerosis (AVS, a precursor of calcification), mitral annulus calcification (MAC) or both.

Generally, aortic valve sclerosis (AVS)/calcification (AVC) is defined as the presence of sclerotic and calcific lesions that reside within the aortic valve leaflets, not involving the aortic annulus (or coronary artery ostia. Valve sclerosis cannot always be differentiated by ultrasound from calcification except that the latter tends to be whiter in appearance. Specifically, AVC is diagnosed by ultrasound if there is hyperreflectivity of the valve cusps (usually the presence of nodular brightness), with or without obstruction to the outflow (Figure 1). Detection by CT is more precise; indeed, calcification is diagnosed [40] if there are at least 3 contiguous pixels of at least 130 Hounsfield units of brightness. This allows accurate quantitation, typically employing the Agatston score technique [41].

MAC is a chronic, degenerative process of the fibrous support structure of the mitral valve, which most often affects the posterior annulus [42]. MAC is visualized by echocardiography as an echo-dense structure with an irregular, lumpy appearance and associated acoustic shadowing [43] (Figure 2). MAC can also be diagnosed and quantified by CT using the Agatston score technique.

The question of whether MAC and AVC are expressions of atherosclerotic disease or simply reflect a primary degenerative process progressing with advancing age was partially clarified in the late 1980s and 1990s. Several authors [44,45] reported that cardiac calcification and vascular atherosclerosis have many shared risk factors, and thus, the former should be regarded as a manifestation of generalized atherosclerosis. Like atherosclerosis, cardiac calcifications progress over time. Some studies describe an increase in the extent of MAC [46] and AVC [47], which can lead to clinically relevant mitral regurgitation [48], nonrheumatic mitral stenosis [49] and aortic valve regurgitation/stenosis [47]. It is therefore not surprising that more than one author has proposed that AVC and MAC could represent a surrogate marker for underlying atherosclerotic disease and an “easy-to-see” ultrasound window of what is occurring in the arterial beds [50,51].

## 3. Clinical Implications of Cardiac Calcifications

The clinical utility of AVC and MAC detected by noninvasive imaging is supported by a rich literature that highlights the prognostic impact of these two entities. In fact, a landmark paper published in 1999 that analyzed data from the Cardiovascular Health Study showed that the presence of AVS/AVC was associated with near double the risk of all-cause mortality and cardiovascular death, as well as an increased number of atherosclerotic events, including myocardial infarction (MI), stroke and heart failure over a follow-up period of five years (the increased risk remained after adjustment for confounders) [52]. These findings were later supported by the results of a prospective analysis of 6685 participants in MESA [53]. In a follow-up of six years, after adjusting for demographics and cardiovascular risk factors, subjects with AVC had higher risks of cardiovascular (HR: 1.50; 95% confidence interval (CI): 1.10 to 2.03) and coronary (HR: 1.72; 95% CI: 1.19 to 2.49) events compared to those without AVC [53]. A possible explanation is provided by a MESA [54] analysis examining the relationship between AVC and CACs severity. The investigators observed that the prevalence of AVC increased as CACs increased. The prevalence ratio of AVC among patients with mild CACs (1–99) was 1.83 (95% CI, 1.45–2.31) and increased to 3.36 (95% CI, 2.56–4.42) for a CAC > 400 when compared to those with CAC = 0 [54]. To explore if AVC adds predictive value beyond CAC, Blaha et al. [55] described the association of AVC with mortality before and after adjustment for CACs. In a cohort of 8401 asymptomatic middle-age subjects free of known CVD, AVC predicted all-cause mortality independent of traditional CV risk factors and added prognostic information in patients with CACs ≥ 100 [55]. Furthermore, supporting the fact that AVC may be an atherosclerotic mirror of vessels, AVS has been shown to be a powerful independent predictor of carotid atherosclerosis; its presence almost doubles the risk of having carotid plaques [51].

Similar findings are available regarding MAC. Fox et al. [56], in the Framingham Heart Study, described an association between MAC and both fatal (HR: 1.6; 95% CI: 1.1 to 2.3) and nonfatal CV events (HR: 1.5; 95% CI: 1.1 to 2.0), even after adjustment for conventional cardiovascular risk factors. Similar to AVC, studies have found an association between MAC and CAD [57], aortic atheroma [58] and carotid plaque [59]. Data from MESA have again provided insight. In an analysis of ~7000 patients MAC, was independently associated with increasing CACs in all racial groups [60].

More than a meta-analysis [61,62] published in recent years confirms the negative prognostic impact of cardiac calcifications. According to the pooled analyses of published studies (including 44,000 patients) conducted by Pradelli et al. [63], risks for all-cause mortality, cardiovascular mortality and CAD were 14%, 53% and 150% higher among subjects with AVC than in those without. Likewise, subjects with MAC had increased risk for all-cause mortality, cardiovascular mortality, stroke and MI of 53%, 65%, 42% and 40%, respectively, versus those without. Furthermore, patients with cardiac valve calcifications are characterized by a worse functional and hemodynamic profile compared to patients with a normal valve. The presence of valve calcification is associated with diastolic dysfunction (higher left ventricular filling pressure) and higher pulmonary artery systolic pressure, independently of a variety of clinical and Doppler echocardiographic parameters [64]. Finally, among certain subgroups of patients (chronic renal failure, diabetes, rheumatoid arthritis), the presence of cardiac calcification has been shown to have an important value in identifying those at increased risk of future CV events [39,65].

Thus, the recognition of even small calcium deposits on valves and other cardiac structures is clinically relevant, both as a marker of systemic atherosclerosis and as a predictor of cardiovascular events. At this point, it is important to note that there is no proven method to reduce the burden of cardiac calcification [66]. The progression of valve disease (i.e., aortic stenosis) is not slowed down by any pharmacologic intervention (e.g., statins); accordingly, the serial assessment of valve calcification is not useful to evaluate therapeutic response. Nevertheless, CV events are reduced in patients with calcific aortic stenosis treated with lipid-lowering drugs [67].

## 4. How to Detect Cardiac Calcification?

### 4.1. Ultrasound

Currently, there are no standardized echocardiographic methods, similar to the Agatston coronary calcium score obtained on CT, to quantitatively assess the extent of cardiac calcification. However, conventional ultrasound imaging, which is widely available, low cost and radiation-free, can provide a semiquantitative assessment of this disease process [68]. Early studies simply marked the presence or absence of MAC [57] and AVC [52]. In recent years various scores or algorithms that can semiquantitatively assess the extent of noncoronary heart calcium have been proposed. The importance of such scores lies in the fact that findings of AVC, MAC and calcification of the ascending aorta, detected by standard 2D-transthoracic and transesophageal echocardiography, are associated not only with coronary and total cardiac calcium burden as assessed by CT but also with the presence of CAD by angiography [69] and, most importantly, with hard CV outcomes [70]. Proposed scores have generally examined the number and extent of calcifications in four sites: aortic valve, mitral annulus, ascending aorta and papillary muscle [71] [72]. Tolstrup et al. [73] showed that the presence of AVC, MAC and aortic root sclerosis (ARS), identified by transesophageal echocardiography, were highly associated with aortic atheromatous disease and CV disease in men. In this study (Table 1), each aortic cusp was scored from 0 (normal) to 4 (severely sclerotic/calcified); the mitral annulus was graded as normal, mildly (<5 mm), moderately (5 to 10 mm) and severely calcified (>10 mm), and ARS was considered present when the anterior or posterior wall showed an increased echo reflectance and thickness >2.2 mm. The ascending, transverse and descending aorta were scored from normal to severely affected by calcification (Grades 0 to 3).

A semiquantitative echocardiographic algorithm (Table 1) using simple transthoracic echocardiographic parameters (MAC, AVC, ARS), called the Calcium Score Index (CSI), was later developed by Corciu et al. [74]. ARS was defined following the criteria proposed by Tolstrup et al. [73]. AVS/AVC was characterized as an enhanced echogenicity and thickening of the cusps or the presence of calcifications, while MAC was defined as an intense echo-producing structure located at the junction of the atrioventricular groove and posterior mitral leaflet and quantified from mild to severe, considering its thickness and length. The CSI range was from 0 (normal) to 10 (diffuse calcification of the aortic root, valve and mitral annulus), and a CSI > 4 was associated with a 72% chance of having a medium to high FRS, a 77% chance of having CAD and a 61% chance of having an abnormal left ventricular mass index.

Gaibazzi et al. [72] demonstrated that the CSI efficiently risk-stratified a clinical sample referred for stress echocardiography. Using a CSI cut-off > 0 (i.e., calcium was present in ≥1 site), the authors observed a sensitivity and specificity of 74% and 60%, respectively, and an area under the curve of 0.67) for the prediction of death/MI. Moreover, the addition of the CSI to clinical and wall motion data resulted in the substantial reclassification of patient risk (NRI: 57%). Interestingly, the best CSI cut-off to predict cardiac events was 0, which is probably the easiest score to apply clinically, because accordingly, CSI is “abnormal” whatever amount of calcium is present in at least 1 of the predetermined cardiac sites. Similarly, the APRES study [75] found ultrasound assessment of calcification in patients with no known CAD to be an independent predictor of angiographic CAD, which added incremental value to the FRS across all pretest risk strata (low, intermedium, high). Again, the addition of the CSI resulted in the significant reclassification of patient risk (NRI 53%), especially in the identification of severe CAD, i.e., >70% stenosis of at least one coronary artery (NRI 70%). Thus, such an echo score may potentially refine clinical selection for invasive diagnostic procedures as conventional coronary angiography; for example, suggesting to “downgrade” it to a noninvasive CT angiography if such an ultrasound “marker” appears normal. Interestingly, the utility of the noninvasive assessment of cardiac calcification has also proved useful in patients with established CAD [76]. In the Heart and Soul Study, a composite calcium score > 2 had a 2.5-fold increased risk of death compared to a score of 0, confirming that a simple echocardiographic cardiac calcium score is an independent predictor of death in secondary prevention patients. Pressman et al. [77] described a “global” cardiac calcification echo score (Table 1), including a semiquantitative assessment of calcifications in the aortic valve and root, the mitral valve and annulus and the submitral apparatus, and found it correlated well with CACs assessed by CT. Specifically, an echocardiographic calcium score of ≥5 had a sensitivity and specificity of 43% and 85% for detection of CACs > 400, with an area under the curve of 0.761. This echo score is able to refine risk as defined by FRS or pooled cohort equations, and it appears to be particularly useful in low-risk groups where it can identify those at increased risk who might otherwise be missed by the traditional risk models [78]. Furthermore, Pressman’s echo score was shown to be a significant predictor of stroke and mortality [79].

A new echocardiographic calcification score “echoCCS” [80,81] (Table 1) proved its value as an independent predictor for significant CAD and for all-cause mortality in patients with a high and low/intermediate CV risk profile. The main difference from the other scores is that a fifth cardiac structure, the ventricular septum, is systematically evaluated for the presence or absence of calcification beyond the aortic valve, the aortic root, the mitral annulus and the papillary muscles. Adding a point for calcification of the ventricular septum makes no difference in the final predictive ability of the score, but this scoring system appears easier to calculate than the others. Indeed, it is calculated according to the presence (1 point) or absence (0 points) of any calcifications among the above-mentioned structures with a final score in the range from 0 to 5.

The need for a true quantitative assessment of cardiac calcification by echocardiography led to the development of a new technique that uses a multipulse scheme termed “eSCAR” [82]. It has been demonstrated that normal myocardium has a “linear” response to ultrasound interrogation, while calcified tissues produce a prominent “nonlinear” response. Thus, eSCAR is able to quantify cardiac calcification due to the high signal amplitude ratio between calcified and noncalcified tissues. This approach needs more data to confirm its clinical validity but represents an interesting and potentially useful technique to quantify calcium deposits using echocardiography. Finally, three-dimensional (3D) echocardiography may allow the more accurate assessment of thickness and contour of plaque or calcifications, which are markers of the atherosclerotic burden of the vascular tree as in the coronary and peripheral arteries [83] (Figure 3 and Appendix A). However, this technique is currently limited by poor resolution compared to 2D imaging.

### 4.2. Comparison between Techniques: Ultrasound vs. Cardiac CT

It should be emphasized that the only validated imaging technique capable of quantifying noncoronary cardiac calcium is gated CT. The extent of cardiac calcification on CT can be expressed by applying the same Agatston [41] score used for assessing the CACs. In this way, it is possible to quantify the calcium score of various cardiac structures (aortic valve, mitral, papillary, aortic arch, septum) without the need for contrast agents. This allows to evaluate, through a single radiological examination, the amount of total cardiac calcium, both coronary and noncoronary.

Therefore, after having already addressed and clarified both the intrinsic limitations of cardiac CT (both for CACs and for noncoronary calcium assessment) and the clinical utility of the detection of cardiac calcifications, it remains to be established whether the above-mentioned ultrasound scores correlate adequately with the “real” cardiac calcium quantified by the CT. Therefore, while cardiac CT remains the reference method to quantify coronary and noncoronary cardiac calcium, the use of echocardiography for this purpose, if validated against CT, would portend the inherent advantages of low cost, portability and radiation safety. The first study that tried to answer this question was that of Pressman et al. [77]. The authors demonstrated that, in a small population of 41 patients, comparing the total echocardiographic calcium score with the CT noncoronary calcium score yielded a positive and significant correlation (rho = 0.56). Comparing the total echocardiographic calcium score with the CACs also showed a significant, but less strong, association (rho = 0.46). Supporting these favorable results are data from 141 subjects analyzed by Gaibazzi et al. [68]. In fact, it has been found that a calcium score easily obtainable during standard echocardiography, is moderately to strongly (Spearman’s rho = 0.64) correlated with noncoronary calcium score by CT and that the two techniques had similar area under the curve (AUC = 0.77) for the prediction of severe CACs (>400). While this study only found a moderate correlation between the echo score and CACs, the study conducted by Hirschberg et al. [80] observed a good Spearman’s correlation (0.73). This means that about 50% of the variability of the CACs depends on the variability of the echo score. Further studies, possibly with a larger sample size are needed to clearly identify the true relationship between the semi-quantitative ultrasound method and the quantitative one by CT.

Some intrinsic limitations of classical ultrasound need to be highlighted. Primarily, ultrasound is an operator-dependent imaging technique presenting unique challenges, such as low imaging quality caused by noise and artifacts. To interpret correctly the information acquired by the scan, clinical experience is required. Despite these inherent limitations to quantification by echocardiography, multiple studies have found the various echocardiographic calcium scores to have good reproducibility, with acceptable intra- and interobserver variability [72,77,81]. Moreover, the echocardiographic detection of calcium relies on the ability of ultrasound to identify even a minimal quantity of calcium; therefore, increased echolucency caused by fibrosis might be misinterpreted as calcification. Finally, before wider use of ultrasound calcium scores in all echocardiographic laboratories could be possible, a rigorous and objective technique’s standardization is required. The application of artificial intelligence (especially deep learning) to ultrasound could allow advanced automatic image analysis potentially useful to assist in ultrasound evaluations and/or to make such assessments more objective and accurate [84].

## 5. Conclusions

In conclusion, the scientific evidence suggests that the detection of cardiac calcifications by ultrasound represents a promising tool for identifying subclinical atherosclerosis and improving risk stratification in asymptomatic subjects. Echocardiography provides an easy, inexpensive and “bedside” tool for the reclassification of subjects into higher- or lower-CV-risk group categories, leading to the more appropriate application of preventive treatments and more appropriate use of diagnostic tests for identifying significant coronary or extracoronary artery stenosis. An appropriately designed and powered prospective study would appear to be the next step in the investigation of this promising method of detecting subclinical atherosclerosis.

## Figures and Tables

**Figure 1 jcm-10-01668-f001:**
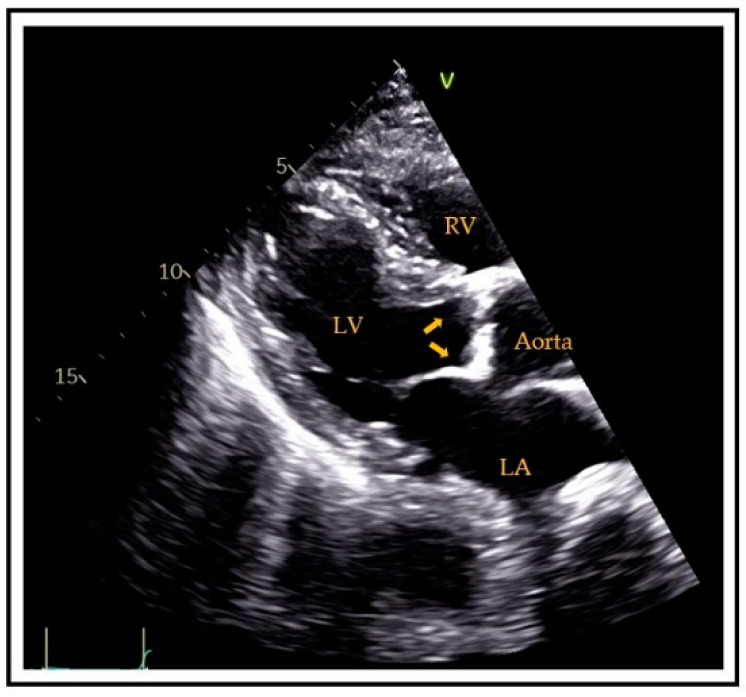
Transthoracic parasternal long-axis view showing aortic valve calcifications (arrows). RV = right ventricle, LV = left ventricle, LA = left atrium.

**Figure 2 jcm-10-01668-f002:**
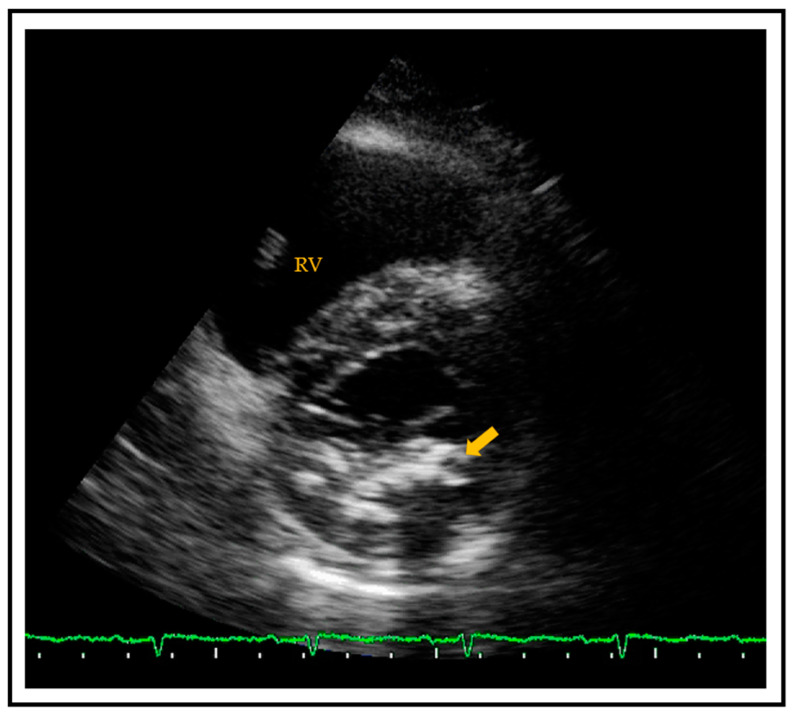
Transthoracic parasternal short axis view showing a calcified mitral posterior annulus (arrow). RV = right ventricle.

**Figure 3 jcm-10-01668-f003:**
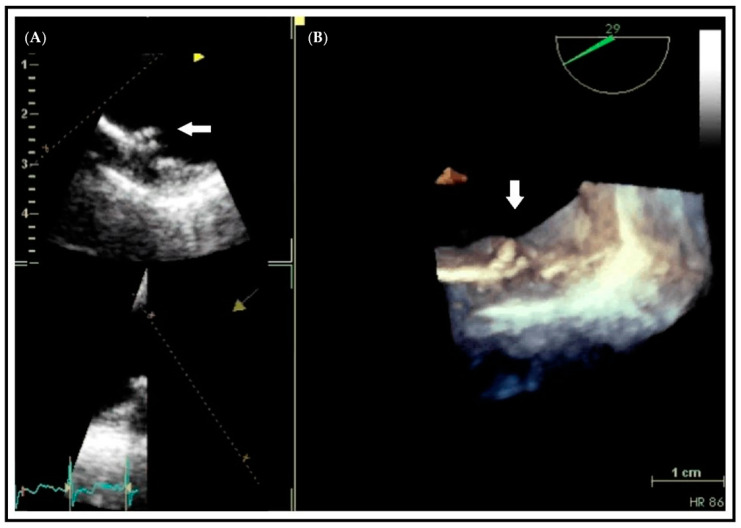
Transesophageal echocardiographic study of a protruding plaque in the descending aorta (arrow). Differences in the visual assessment of aortic atherosclerotic plaque between a 2D-based image (**A**) compared with the 3D-based image (**B**). The entire contour of the complex plaque is seen in the 3D image.

**Table 1 jcm-10-01668-t001:** Different Ultrasound Cardiac Calcification Grading Score. (**A**) Cardiac calcification grading score proposed by Tolstrup et al. [73]. (**B**) Semiquantitative Calcium Score Index (CSI) proposed by Corciu et al. [74] and modified and adopted by Gaibazzi et al. [72]. (**C**) Global cardiac calcium ultrasound score elaborated by Pressman et al. [77]. (**D**) Echocardiographic calcification score “echoCCS” proposed by Hirschberg et al. [80].

**(A) Tolstrup et al. [73]**	**Grade**	**Aortic Valve Sclerosis**		**Mitral Annulus Calcification**		**Aortic Atheromatous Disease**			
Final Score Range: 0–10	0	No thickening		Normal		Normal, <2 mm intimal thickness			
	1	Slight ↑ reflectance, thickness < 2 mm		<5 mm		Intimal thickness 2–4 mm			
	2	Mild ↑ reflectance, thickness 2–4 mm		5–10 mm		Intimal thickness > 4 mm			
	3	Generalized hyper-reflectance, thickness > 4 mm		>10 mm		Any protruding/mobile plaque			
	4	Markedly hyper-reflectant masses, thickness > 6 mm							
**(B) Corciu et al. [74]**	**Grade**	**Papillary Muscle Calcium**	**Mitral Annular Calcium**	**Aortic Valve Sclerosis**		**Aorta Root Calcium**			
Final Score Range: 0–8	0	Absent	Absent	Normal cusp thickness (<2mm) and normal reflectivity		Absent			
	1	Present	<5 mm	Cusp thickness > 2 mm and/or increased reflectivity		Present			
	2		5–10 mm	Thickness > 4 mm and/or diffuse or focal cusp hyperreflectivity					
	3		>10 mm	Thickness > 6mm and/or marked echoreflectivity					
**(C) Pressman et al. [77]**	**Grade**	**Posterior Annulus Calcium**	**Posterior Mitral Leaflet Restriction**	**Anterior Annulus Calcium**	**Anterior Mitral Leaflet Restriction**	**Mitral Valve Calcium**	**Subvalvular Apparatus Calcium**	**Aortic Valve Calcium**	**Aortic Root Calcium**
Final Score Range: 0–13	0	Absent	Absent	Absent	Absent	Absent	Absent	Absent	Absent
	1	1/3 of annulus	Present	Present (any mobility reduction)	Present (valve opening <10mm)	Mild	Present	Nodules in <3 leaflets	Present
	2	2/3 of annulus				>Mild		Nodules in 3 leaflets but non-restrictive	
	3	3/3 of annulus						Restrictive (meam gradient > 15 mmHg or reduced leaflets motion)	
**(D) Hirschberg et al. [80]**	**Grade**	**Papillary Muscle Calcium**		**Mitral Annular Calcium**		**Aortic Valve Sclerosis**		**Aorta Root Calcium**	**Septal Calcium**
Final Score Range: 0–5	0	Absent		Absent		Absent		Absent	Absent
	1	Present		Present		Present		Absent	Present

## Data Availability

No new data were created or analyzed in this study. Data sharing is not applicable to this article.

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
