# Peer review of "Cardiovascular Calcification as a Marker of Increased Cardiovascular Risk and a Surrogate for Subclinical Atherosclerosis: Role of Echocardiography"

_jcm, 2021, doi:10.3390/jcm10081668_

Round 1

Reviewer 1 Report

This review on the use of ultrasound cardiac calcification in assessing risk stratification and predicting coronary calcification comprehensive, very well written and provides a strong foundation for future prospective research in the field. The manuscript needs thorough English checking though.

Author Response

This review on the use of ultrasound cardiac calcification in assessing risk stratification and predicting coronary calcification comprehensive, very well written and provides a strong foundation for future prospective research in the field. The manuscript needs thorough English checking though.

Thank you reviewer. A careful English language check has been carried out.

Reviewer 2 Report

Dear Authors

That is a well-written review summarizing the changing views on the role of imaging techniques on cardiovascular risk assessment.

The discussion about the additional merit   of imaging techniques for adequate classification of risk is essential and has didactic potential for  readers.

All visual parameters dealing with calcification having an impact on cardiovascular risk are not exactly a visualization of the atherosclerotic process but can  only be treated as a surrogate marker for possible atherosclerosis .    Extensive calcifications can be also linked to other pathophysiological situations ( eg. hypercalcemia and others).

Authors present and promote the use of calcium scores from multiple  locations calculated during echcardiograophic examination and its correlation with a “gold standard” CT with Agatson scale. Despite the potential  of the use of calcification recognition by echocardiography, some drawbacks are in my opinion not fully addressed. In ultrasound examinations (vascular or cardiac), the characterization of echolucency and  reflectance  is highly dependent on the setting and type of the ultrasound apparatus and  the type of  probe used. Therefore, the results between different ultrasound laboratories are not fully comparable. The tool like Agatson scale used in CT examinations  in ultrasound unfortunately is not standardized. This is not always  fully appreciated and should be clearly stated in your review.

My main concern

  1. Title – Cardiac Valve Calcification as a Marker of Subclinical Atherosclerosis: The role of echocardiography- in the present form can be misleading -it does not reflect fully  subclinical atherosclerosis but rather a marker of increased cardiovascular risk.

Please consider changing the title to : Cardiac Valve Calcification as a Marker of Increased Cardiovascular Risk and a surrogate for  Subclinical Atherosclerosis

  1. Please add some additional paragraphs on the limitations of classical ultrasound for the characterization of tissue composition and the need for standardization before wider use in all echocardiographic laboratories.

Besr regards,

Author Response

-Question 1) My main concern: Title – Cardiac Valve Calcification as a Marker of Subclinical Atherosclerosis: The role of echocardiography- in the present form can be misleading -it does not reflect fully  subclinical atherosclerosis but rather a marker of increased cardiovascular risk. Please consider changing the title to : Cardiac Valve Calcification as a Marker of Increased Cardiovascular Risk and a surrogate for  Subclinical Atherosclerosis

-Answer 1) Thank you Reviewer for your comments and suggestions. Since in the manuscript we discuss not only valve calcifications but also those of the mitral annulus, aortic root, carotids and coronaries (indirectly),  we would like to propose this title:

 “Cardiovascular Calcification as a Marker of Increased Cardiovascular Risk and a surrogate for  Subclinical Atherosclerosis: Role of Echocardiography”.

 Do you think it can be clearer and more appreciable?

-Question 2) Please add some additional paragraphs on the limitations of classical ultrasound for the characterization of tissue composition and the need for standardization before wider use in all echocardiographic laboratories.

-Answer 2) Thank you Reviewer. We have implemented the manuscript with the following paragraph:

“Some intrinsic limitations of classical ultrasound need to be highlighted. Primarily, ultrasound is an operator-dependent imaging technique presenting unique challenges, such as low imaging quality caused by noise and artifacts; to interpret correctly the information acquired by the scan, clinical experience is required. Despite these inherent limitations to quantification by echocardiography, multiple studies have found the various echocardiographic calcium scores to have good reproducibility, with acceptable intra and inter-observer variability [73] [78] [82]. Moreover, the echocardiographic detection of calcium relies on the ability of ultrasound to identify even a minimal quantity of calcium; therefore, increased echolucency caused by fibrosis might be misinterpreted as calcification. Finally, before wider use of ultrasound calcium scores in all echocardiographic laboratories could be possible, a rigorous and objective technique's standardization is required. The application of artificial intelligence (especially deep learning)  to ultrasound could allow advanced automatic image analysis potentially useful to assist in ultrasound evaluations and / or to make such assessments more objective and accurate”.